# Action Mechanisms of Effectors in Plant-Pathogen Interaction

**DOI:** 10.3390/ijms23126758

**Published:** 2022-06-17

**Authors:** Shiyi Zhang, Cong Li, Jinping Si, Zhigang Han, Donghong Chen

**Affiliations:** State Key Laboratory of Subtropical Silviculture, College of Forestry and Biotechnology, Zhejiang A&F University, Hangzhou 311300, China; zhangshiyi_1997@163.com (S.Z.); congli@zafu.edu.cn (C.L.); lssjp@163.com (J.S.)

**Keywords:** pathogen, effector, plant immunity, virulence promotion, infestation process

## Abstract

Plant pathogens are one of the main factors hindering the breeding of cash crops. Pathogens, including oomycetes, fungus, and bacteria, secrete effectors as invasion weapons to successfully invade and propagate in host plants. Here, we review recent advances made in the field of plant-pathogen interaction models and the action mechanisms of phytopathogenic effectors. The review illustrates how effectors from different species use similar and distinct strategies to infect host plants. We classify the main action mechanisms of effectors in plant-pathogen interactions according to the infestation process: targeting physical barriers for disruption, creating conditions conducive to infestation, protecting or masking themselves, interfering with host cell physiological activity, and manipulating plant downstream immune responses. The investigation of the functioning of plant pathogen effectors contributes to improved understanding of the molecular mechanisms of plant-pathogen interactions. This understanding has important theoretical value and is of practical significance in plant pathology and disease resistance genetics and breeding.

## 1. Introduction

Since the inception of plant-pathogen interaction, a fierce but silent game has been played between pathogens and plants [1], where, on the one hand, pathogenic microorganisms plunder nutrients from host cells for survival and reproduction, and, on the other hand, host plants employ various defense strategies to inhibit pathogen growth [2]. Initially, plants used pattern recognition receptors (PRRs) to recognize pathogen-associated molecular patterns (PAMPs) to stimulate pattern-triggered immunity (PTI) defense responses and thus to limit pathogen growth. In turn, as pathogens evolved effectors that inhibited the first layer of immunity, plants evolved nucleotide-binding domain and leucine-rich repeat protein (NLR) receptors that recognized the effectors, thus leading to a second layer of immune effector-triggered immunity (ETI) [3,4,5]. In this process, the effectors of pathogens have been shown by many studies to be important pathogenic factors in the process of infesting plants [6]. The effectors act in multiple ways on different targets, suppressing plant immunity, manipulating plant physiology, and being recognized by host defense mechanisms, thus promoting pathogen infestation, expansion, and colonization. In a narrow sense, effectors are proteins secreted by pathogens into the extracellular and intracellular spaces of host plants [7,8,9,10], which are capable of eliciting effector-triggered plant immunity. The specific molecular characteristics of a typical effector are as follows: approximately 50–300 amino acid residues in length, containing an N-terminal signal peptide with a highly specific sequence, with no transmembrane structural domain, no anchor site for glycosylphosphatidylinositol (GPI), no subcellular localization signal for mitochondria or other intracellular organelles, and being rich in cysteine residues [11,12]. The mechanisms of action of secreted proteins are divided into two categories: direct (pathogen itself) and indirect (natural host resistance) [13,14,15]. However, there are also effector proteins that do not have a signal peptide structure which are still able to secrete extracellularly, having characteristics of unconventionally secreted proteins [16]. Although typical effectors mostly comprise small molecule secreted proteins, certain non-protein-like molecules, such as secondary metabolites [17], toxins [18], phytohormones [19], and sRNAs [20] have been found to be secreted extracellularly by various pathogens and to exhibit similar effects as effectors [21]. Therefore, a broad definition of effectors has been proposed, i.e., “proteins and small molecules that alter the structure and function of host cells, thereby promoting the colonization of pathogens” [22].

To date, type III effectors in bacteria [23,24], LysM effectors in fungus [25], and RxLR effectors in oomycetes [26,27] have been extensively studied and many details about their biological functions are known so that the mechanism of action of effectors is gradually being unveiled. Ray et al. [28] have provided a systematic review of the interaction between virus effectors and plants, so we do not repeat these details further here. Previously, Tariqjaveed et al. [29] conducted a comprehensive review of fungal effectors based on their targets of action, and Pradhan et al. [30] explored the double-edged role of fungal effectors in terms of different trophic types. While previous studies have focused more on discussing the spatial targets of effectors and the differences in effector deployment strategies among different pathogens, this review focuses on the role of effectors in different biological processes during infestation, highlighting recent progress made in the study of plant-pathogen interaction models and phytopathogenic bacterial, oomycete, and fungal effectors, and looking forward to the future challenges of effector research. 

## 2. Plant-Pathogen Interaction Models

Since Flor proposed the “gene-for-gene” hypothesis between pathogen-free genes and plant disease resistance genes in 1942 [31], the understanding of plant immunity mechanisms has been further developed. Models proposed to represent plant-pathogen interaction include the guard model, which considers indirect interactions between plant R genes and pathogen AVR [32], the zigzag model, which proposes dynamic changes in plant-pathogen interactions [3,4], the “decoy” hypothesis, which explains the existence of functionally redundant R genes and non-functional effectors [33,34], and the iceberg model [35], which proposes crosstalk between intercrossing units. Interaction models between plants and pathogens have been continuously refined and new ideas have emerged.

The zigzag model, known as the “central dogma” in plant pathology, is the most accepted and recognized model of interactions. The model suggests that plant innate immunity is composed of two main components, the pathogen-associated molecular pattern PAMP-primed immune response (PTI) and the effector-primed immune response (ETI) (Figure 1). The first line of defense is the PTI response caused by the recognition of pathogen-derived molecules PAMP by pattern recognition receptors (PRRs) on the cell surface, mainly receptor-like kinases (RLKs) and receptor-like proteins (RLPs). The main PAMPs that are currently well studied are flagellin flg22 [36,37,38], elongation factor EF-Tu [39,40], some lipopolysaccharides and oligosaccharides [41,42] in bacteria, chitin and cell wall derivative glucan in fungus, and glutamine aminotransferase and β-glucan [43] in oomycetes. This pattern recognition is usually associated with calcium influx, callose deposition, reactive oxygen species (ROS) burst, miRNA pathway activation, MAPK cascade activation, and induced expression of a large number of defense-related genes, such as disease-process-related protein (PR) [12,44,45]. To resist PTI, pathogens secrete effectors into host plant cells to avoid host recognition as well as to inhibit host defense responses to promote pathogenesis [6,11,46,47,48]. This initiates a second line of plant immune defense, ETI, in which NLRs encoded by R genes in plants recognize effectors secreted by pathogens, leading to ETI [12,49]. One of the characteristics of ETI is the induction of an allergic hypersensitive response (HR), leading to rapid cell death at the site of pathogen attack and limiting the spread of the pathogen [50]. In addition, a series of signal transduction events, such as NO, lipids and various phytohormones [3,5] that sense pathogen attack, are triggered to activate downstream plant immune defense responses. Recent studies have found that PTI and ETI are not independent immune signaling pathways. PTI is an integral component of ETI during pathogen infection—activation of ETI enhances the PTI signaling pathway, but ETI by itself is not sufficient to completely activate plant resistance, and ETI probably functions by directly co-opting PTI anti-pathogen mechanisms [51,52,53].

## 3. Action Mechanisms of Effectors in Plant-Pathogen Interaction of Pathogen Effectors

Plant pathogens produce distinct infestation strategies based on their different nutritional modes [54,55]. Biotrophic pathogens must manipulate host physiological activities to obtain nutrients from living host cells and tissues in order to survive and complete their life cycle, and, therefore, secrete effectors to suppress host immunity while minimizing damage to host cells so that the pathogen can colonize living cells [54,56]. In contrast, necrotrophic pathogens acquire nutrients from dead cells [57] and, therefore, secrete effectors to induce host cell necrosis when they proliferate on plant tissues [22,56]. The infestation process of hemibiotrophic pathogens such as *Bipolaris sorokiniana*, *Verticillium dahlia* and *Magnaporthe oryzae* is more like a combination of both, where the pathogens selectively secrete different effectors at different spatial and temporal levels. They suppress the natural immunity of the plant during the biotrophic phase and induce non-specific cell death during the necrotic phase [22,54].

Once secreted extracellularly by pathogens, effectors are transferred to different parts of the host cell to interfere with various biological processes. The extraplasmic effectors dismantle the host physical and chemical barriers and break the first line of defense of the plant. The intracellular effectors target essential immune components in the plasma membrane, cytoplasm (including organelles) and nucleus, suppress host immunity, reprogram host physiology and favor pathogen colonization [29]. In the following sections, we synthesize the effectors that are well-researched under each mechanism for further classification and summary. Based on existing research results, these are emphasized in different sections.

### 3.1. Breaking down Physical Barriers

#### 3.1.1. Manipulation of Host Plant Stomatal Defenses

Stomata are the main point of entry of many pathogens into plants. However, guard cells are active immune sensing cells that, once they detect the characteristics of microorganisms, can rapidly cause stomatal closure, thus preventing the entry of pathogens [58]. Bacterial pathogens can overcome stomatal defenses by inhibiting PAMP-triggered stomatal closure or by actively inducing stomatal opening through various mechanisms (Figure 2) [59]. *Pseudomonas syringae* produces a series of effectors to manipulate stomata, thereby enhancing pathogen entry. Immune-mediated stomatal closure requires the phytohormone salicylic acid (SA) and, conversely, the hormone jasmonic acid (JA) has an antagonistic effect on SA [60]. JA signaling is negatively regulated by the JAZ transcriptional repressor [61]. The *P. syringae* effector AvrB interacts with plant RPM1-interacting protein 4 (RIN4) to enhance the activity of plasma membrane H^+^-ATPase AHA1, which promotes stomatal opening by directly regulating guard cell turgor pressure [62]. In addition, RIN4 can interfere with hormone signaling, cause upregulated expression of JA-responsive genes, and enhance plant susceptibility [63], which is achieved by facilitating the interaction between COI1 and JAZ proteins and degrading JAZ proteins [62]. Similarly, the cysteine protease (CP) effector HopX1 induces JA signaling, suppressing stomatal immunity by degrading multiple JAZ transcriptional repressors, leading to the activation of JA-regulated genes [64]. The *Xanthomonas campestris* effector XopS stabilizes the negative regulator WRKY40, reducing the expression of its targets, which include SA response genes and the JA repressor JAZ8, preventing stomatal closure in response to bacterial elicitors [65]. These results all suggest that the JA pathway is a common target for effector regulation of stomatal defense, emphasizing the importance of overcoming stomatal defense for successful colonization by pathogens. In contrast, the *P. syringae* effectors HopM1 and AvrE1 target the ABA signaling pathway, increasing ABA accumulation in guard cells, inducing stomatal closure, and promoting water-soaking lesions. The guard-cell-specific ABA transporter ABCG40 is required for HopM1-induced water-soaking lesions [66].

#### 3.1.2. Degradation of Plant Cell Walls

Many plant pathogens, especially those lacking specific penetrating structures, secrete various types of cell-wall-degrading enzymes (CWDEs) to disrupt host cells and promote colonization, including glycoside hydrolases, glycosyltransferases, and pectin lyases to hydrolyze cell wall components (Figure 2) [67,68]. This mechanism is particularly common in necrotrophic pathogens, with certain CWDEs correlating positively with virulence in *Botrytis cinerea*, *V. dahliae* and *Mycosphaerella graminicola* [69,70,71,72,73].

A fraction of CWDEs target the cuticle on the cell wall, and the waxy cuticle is protective against biotic and abiotic stresses [74,75]. *Colletotrichum gloeosporioides*, the pathogen of oil tea anthracnose, secretes the cutinase CglCUT1, which degrades the cuticle during pathogenesis [76]. Some CWDEs can degrade polysaccharides and cellulose in the cell wall, releasing oligosaccharides as PAMPs to trigger plant immunity [77,78]. *M. oryzae* secretes the endoglucanases MoCel12A and MoCel12B during infection, targeting hemicellulose in the rice cell wall and releasing two specific oligosaccharides, the trisaccharide 3^1^-β-D-cellulosyl glucose and the tetrasaccharide 3^1^-β-D-cellobiose glucose. These oligosaccharides both serve as specific DAMPs (danger-associated molecular patterns) recognized by the OsCERK1 and OsCEBiP immune complexes during rice blast infection. In contrast, CfGH17-1 and CfGH17-5 produced by the tomato leaf mold fungus *Cladosporium fulvum* release DAMPs through degradation of β-1,3-glucan, inducing plant cell death [79]. *B. cinerea* secreted BcGs1, a glucan 1,4-alpha-glucosidase, induces cell death through the same mechanism [80].

#### 3.1.3. Attacking Plasmodesmata–Callose Regulation

Plasmodesmata (PD) are membrane-lined pores that connect adjacent cells and mediate symplast communication in plants. Under immune stimulation, plants shut down PD as part of their immune response, which is also used by pathogens to facilitate their infection of the host. The effector transfer pathway via PD is critical for the successful plant colonization by pathogens [81,82,83].

To counteract PD closure and callose accumulation, effectors can target PD [81,84,85], expanding PD pore size and controlling cytoplasmic continuity. During *Fusarium oxysporum* infection, Avr2 and Six5 effectors interact at PD to enlarge pore size [81]. Six5 plays a role similar to that of viral motor proteins [86] in this process. In the presence of Six5, AVR2 interacts with Six5 and the PD are opened, so that AVR2 can move between cells through the PD. The oomycete pathogen *Phytophthora brassicae* RxLR3 [85] effector inhibits callose accumulation in PD by interacting with the callose synthases CalS1, CalS2, and CalS3. The bacterial pathogen *P. syringae* effector HopO1-1 localizes to PD and regulates PD permeability [87]. Expression of HopO1-1 in *Arabidopsis* increases the distance of PD-dependent molecular flux between adjacent plant cells [87]. In addition, HopO1-1 interacts with and destabilizes the plant PD-localized proteins PDLP7, and possibly PDLP5, whereas bacterial proliferation is significantly increased in mutant plants lacking PDLP7 or PDLP5 [88,89], which indicates that PDLP7 and PDLP5 are likely to be involved in plant immunity against bacteria.

#### 3.1.4. Destruction of the Host Plant Cytoskeleton

The cytoskeleton responds to both abiotic and biotic stimuli [90]. When individual plant cells come into contact with pathogenic microorganisms, the regular cytoskeletal structure undergoes rapid changes, locally transporting the cargo used for defense execution [91].

Some effectors manipulate host physiological and metabolic processes by disrupting the formation of the host cytoskeleton (Figure 2). The *P. syringae* type III effector HopW1 inhibits endocytosis as well as the transport of certain proteins to the vesicle by forming a complex with actin and disrupting the actin cytoskeleton. The C-terminal region of HopW1 shortens the length of actin filaments, thereby lysing in vitro F-actin in vitro [92]. In contrast, *Blumeria graminis* f.sp. *hordei* (Bgh) ROPIP targets the barley ROP GTPase HvRACB and manipulates host cell microtubule organization to facilitate its own cell entry [93]. Interestingly, *Xanthomonas oleifera* T3E XopR also undergoes liquid–liquid phase separation (LLPS) by hijacking the multivalent intrinsically disordered region IDR-mediated interactions of the *Arabidopsis* actin cytoskeleton. XopR gradually translocates into the host cell during infection and forms macromolecular complexes with actin-binding proteins in the cell cortex, progressively manipulating multiple steps of actin assembly and disrupting the host actin cytoskeleton [94].

### 3.2. Creating Conditions Favorable to Infestation

#### 3.2.1. Construction of Hydrophobic Space

Mycelium or spores often need to escape from the aqueous environment and grow in air when infesting plants, and some effectors promote infestation by constructing hydrophobic spaces between pathogen-host plants. Hydrophobins [95] (HP) are secreted small-molecule fungal proteins with eight conserved cysteine residues [96,97,98] that self-assemble into amphiphilic monomolecular membranes at the hydrophilic–hydrophobic interface. They provide a hydrophobic protein coating for mycelium or spores that may be involved in attachment to hydrophobic surfaces, interactions with the environment, and the host defense system and other processes that contribute to spore dispersal and aerial growth of mycelium during escape from aqueous environments [97,99].

The hydrophobin MPG in *M. oryzae* is highly induced in the early stages of infection [100] and acts as a sensor on the hydrophobic plant surface, triggering the development of attachments [101]. *Mpg1* deletion mutants exhibit reduced virulence due to attachment defects on hydrophobic surfaces and subsequent defects in attachment formation. A surface coating of hyphae with MPG1 has been shown to allow efficient recruitment and retention of cutinase 2, which contributes to appressorium differentiation and penetration (Tanaka 122). The genome of *Fusarium graminearum* also contains four class I hydrophobin genes (FgHyd1~FgHyd4) and one class II gene (FgHyd5) [102]. In mutants carrying single gene deletions, *ΔFgHyd2*, *ΔFgHyd3*, and *ΔFgHyd4* exhibited reduced virulence, which was attributed to the reduced ability of hyphae to penetrate the water-air interface and the reduced attachment of fungal cells to hydrophobic plant surfaces [102]. In addition, the extracellular matrix protein EMP1 of the rice blast fungus [103] also plays a role similar to hydrophobin. An EMP1 knockout mutant had significantly reduced appressorium formation and pathogenicity, but there was no effect on its mycelium growth and sporulation, indicating that EMP1 plays an important role in the appressorium formation of *M. oryzae*.

#### 3.2.2. Induction of Extracellular Alkalinization

Environmental pH plays a powerful role in regulating the growth and development of pathogenic fungus [104] and is a key factor in controlling fungal pathogenicity [105]. Fungal-induced plant infection is usually accompanied by elevated pH in the surrounding host tissues and this extracellular alkalinization is thought to be associated with fungal pathogenesis [106,107].

Rapid alkalinization factor (Ralf) homologs, which are widely present in fungus, can cause an increase in extracellular pH and promote invasive fungal growth by stimulating the phosphorylation of conserved mitogen-activated protein kinases essential for pathogenesis (Figure 2). Endogenous Ralf-Feronia signaling in plants leads to inactivation of the plasma membrane H^+^-ATPase AHA2, which inhibits plant cell elongation [108]. The root-infecting fungus *F. oxysporum* is then able to induce alkalinization and cause plant disease using a functional homologue of the plant regulatory peptide Ralf, a peptide hormone capable of increasing the pH of surrounding fruit tissue or rhizosphere by more than two units, respectively, enhancing fungal colonization by increasing the pH of the apoplastic environment [109,110]. *F. oxysporum* mutants lacking a functional Ralf peptide fail to induce host alkalinization and exhibit markedly reduced virulence in tomato plants while eliciting a strong host immune response. F-Ralf appears to target the plant receptor-like kinase Feronia, which also mediates the response to endogenous plant Ralf peptides [111]. *Arabidopsis* plants lacking Feronia, a receptor-like kinase that mediates the Ralf-triggered alkalinization response, exhibit greater resistance to *F. oxysporum*.

### 3.3. Protecting or Masking Themselves

#### 3.3.1. Inhibition of PTI

Degraded fragments of pathogenic bacteria and plant cell walls may be recognized by plant PRR as PAMPs or DAMPs to induce immunity [112]; consequently, pathogens secrete a range of different effectors to suppress PTI. Chitin, a PAMP known to be recognized by lysin motif (LysM) receptors [112,113], is capable of activating PTI. Here, we use chitin as an example to elucidate several different strategies for effector suppression of PAMP-triggered immunity. For chitin, the main means used by pathogens are (i) protection of the mycelium from degradation by plant chitinases, (ii) inhibition of LysM receptor recognition, (iii) isolation and masking of chitin oligosaccharides, (iv) targeting of chitinases for degradation, and (v) modification and transformation of cell wall components (Figure 3) [112,114,115].

Maintaining cell wall integrity is a conserved strategy for pathogens to inhibit PTI (Figure 3). One of the most important mechanisms for this is to secrete chitin-binding lectins, which bind to the chitin layer to protect it from plant chitinases, thereby avoiding the release of free chitin. The tomato leaf mold *C. fulvum* secretes the chitin-binding lectin Avr4 belonging to the CBM14 family [44], and binds to the fungal cell wall through the chitin-binding domain, thereby protecting cell wall integrity from chitinases [116]. Heterologous expression of Avr4 in *Arabidopsis* and tomato masks the chitin, thereby increasing the virulence of several fungal pathogens [117]. The integrity of the chitin-binding domain, which covers the entire length of CfAvr4, is essential for mycelial protection of chitinases [118]. Moreover, functional homologs of CfAvr4 have been identified in several other fungal species [119,120], suggesting that the chitin-binding domain is likely to be somewhat conserved in fungus. In addition, CBM18 and CBM50 (LysM) share a similar mechanism of action. *Verticillium nonalfalfae*, which invades the xylem of plants, secretes an effector VnaChtBP containing the CBM18 domain. VnaChtBP is highly upregulated during colonization, and its protein product binds to the chitin oligomers and protects the fungal mycelium from chitinase. However, knockdown of VnaChtBP did not affect virulence, which was interpreted as a result of redundancy and convergent evolution of the CBM18 effectors [115]. The addition of the RiSLM protein containing the CBM50 domain from *Rhizophagus irregularis* to *Trichoderma* hyphae provides protection against chitinase in vitro and effectively interferes with the chitin-triggered immune response [121]. The conserved secreted protein VdCP1 in *V. dahliae*, a member of the Cerato-platanin protein (CPP) family, exhibits chitin-binding properties that may protect the fungal cell wall from enzymatic degradation [122]. In *M. oryzae*, the α-1,3-glucan synthase gene MgAGS1 is essential for virulence [123,124]. Since α-1,3-glucan is insensitive to plant-produced enzymes, the outer cell wall layer of α-1,3-glucan can act as a shielding molecule against enzymatic degradation [125,126]. *Ustilago maydis* also secretes Sta1 effectors associated with virulence [127]. Although Sta1 lacks a characteristic domain suggestive of carbohydrate-binding, it is specifically present on the surface of *U. maydis* hyphae but cannot attach to budding cells. Overexpression of Sta1 makes the fungal mycelium more sensitive to chitinase and β-glucanase, suggesting a role for Sta1 in maintaining the fungal cell wall. It is speculated that Sta1 may prevent the release of fungal cell wall-derived elicitors and function as a stage-specific stealth molecule.

Targeting receptor-like kinases to avoid recognition is also a common strategy for pathogens to evade PTI (Figure 3). Plasma membrane-bound PRRs recognize PAMPs or DAMPs, thereby activating plant defenses against pathogens. The bacterial effectors AvrPto and AvrPtoB target multiple PRRs, including flagellin-sensing 2 (FLS2) and elongation factor Tu receptor (EFR) [128]. AvrPtoB acts on the *Arabidopsis* LysM receptor kinase CERK1 and blocks all defense responses through this receptor. AvrPtoB ubiquitinates the CERK1 kinase domain in vitro and targets CERK1 for in vivo degradation [129]. Some fungal effectors also act on receptor-like kinases to regulate immunity, such as the core effector NIS1 of *Colletotrichum* spp. [130]. The two homologous effectors CoNIS1 and MoNIS1 of *Colletotrichum orbiculare* and *M. oryzae* inhibit the kinase activity of BIK1 and BIK1-RbohD interactions and suppress their phosphorylation [130], thereby blocking the chitin-induced immune activation.

Some effectors scavenge or sequester free chitinous oligosaccharides in the apoplast [131] to avoid triggering PTI [132], which is usually achieved by LysM effectors that are widespread in the fungus [133,134]. The tomato leaf mold *C. fulvum* effector Ecp6 is a chitin-binding protein containing the LysM domain. The intrachain LysM dimerization of Ecp6 provides an ultra-high affinity binding groove that competes with the host LysM receptor for chitin binding in the apoplast, isolating chitin oligosaccharides to prevent chitin-triggered immunity [132,135]. Similarly, the *M. oryzae* effector MoAa91 is required for surface recognition and inhibition of the chitin-induced plant immune response; MoAa91 is secreted into the apoplast space by competing with the rice immune receptor CEBIP, where it binds to chitin and chitin oligosaccharides, thereby inhibiting the chitin-induced plant immune response [136]. Such chitin-binding effectors also include SLP1 in *M. oryzae*, ChELP1 and ChELP2 in *Colletotrichum higginsianum* [137,138], RsLysM in *Rhizoctonia Solani* [139], TAL6 in *Trichoderma atroviride* [140] and Vd2LysM in *V. dahliae* [114]. Notably, the N-glycosylation of SLP1 is required for its protein stability and chitin affinity [141], while ChELP1 and ChELP2 are also dependent on glycosylation [138], suggesting that glycosylation is usually essential for fungal LysM effectors to obtain ultra-high chitin affinity.

Another interesting mechanism is that effectors directly target chitinase for degradation to avoid chitin-triggered immunity. *F. oxysporum* can secrete metalloprotease FoMep1 and serine protease FoSep1 to cleave tomato chitinase and promote virulence, while *U. maydis* may secrete metalloprotease UmFly1 to truncate maize chitinase ZmChiA. The glycoside hydrolase family 18 (GH18) in the cacao pathogen *Moniliophthora perniciosa* has evolved an enzymatically inactive chitinase MpChi that sequesters immunogenic chitin fragments to inhibit chitin triggering immunity [142]. *V. dahliae* secretes the serine protease SSEP1 to hydrolyze cotton apoplast class IV chitinase Chi28 [131,143]. *F. oxyporum* f.sp. also secretes the serine protease Sep1 and the metalloprotease Mep1, which act synergistically to degrade tomato extracellular chitinase and protect the fungal cell wall from degradation [144]. Furthermore, the extracellular chitinase MoChi1/MoChia1 in *M. oryzae* competes with OsMBL1, a jacalin-related mannose-binding lectin in rice, and binds to chitin to inhibit chitin-triggered host defense [145]. Interestingly, the rice tetrapeptide repeat protein OsTPR1 competitively binds MoChia1, allowing accumulation of free chitin and reestablishing the immune response [146].

In addition to the positive fight against chitinases and the protection of its own mycelium, an interesting mechanism is the conversion or modification of chitin polymers so that they are not recognized by the receptor. Specifically, the composition of the cell wall is changed by converting the chitin to chitosan with the help of a specific chitin deacetylase [95,147,148]. Chitosan is a relatively poor substrate for chitinases and is relatively inactive in immunogenicity, leading to reduced release of chitin oligomers and thus triggering defense [149,150]. In vitro and in vivo the differentiated infestation structure of the wheat stem rust fungus *Puccinia graminis* f. sp. surface-exposed chitin is converted to chitosan [151]. Studies on the broad bean rust fungus *Uromyces fabae* and maize anthracnose pathogen *Colletotrichum graminicola* have been conducted; fluorescence microscopy with fluorescently labeled lectin wheat germ agglutinin (WGA) revealed that the surface of the infected structures formed on the plant cuticle expose chitin, while the structures formed after invasion of the host surface do not expose chitin, but rather have glycosylated modifications of chitin [152,153]. However, it is uncertain whether these chitosan coverings have a shielding function. The polysaccharide deacetylases VdPDA1 and FovPDA from the invasive xylem fungus *V. dahliae* and *Fusium oxyporum* f. sp. were also recently found to play a role in deregulating chitin-triggered immunity. When the genes encoding specifically secreted chitin deacetylases are deleted, virulence is reduced. It is hypothesized that these enzymes deacetylate the chitin oligomers in the apoplast, which reduces their ability to trigger defense responses [148]. It was recently reported that *U**. maydis* contains a family of seven genes for chitin deacetylase (CDA). Six of them encode enzyme-active proteins, which are differentially expressed during colonization and help evade host recognition by modifying chitin to chitosan. These CDA genes play an important role in pathogenic virulence and cell wall integrity.

Several other effectors have been shown to suppress chitin-triggered immunity directly or indirectly, such as sRNAs of *F. graminearum* that target and silence the resistance-related gene for the chitin exciton binding protein TaCEBiP to suppress the defense response in wheat [154], thereby enhancing *F. graminearum* invasion; the small cysteine-rich effector SCRE2 in *Ustilaginoidea virens* and the lipase-like effector AGLIP1 in *Rhizoctonia solani* have also been shown to impede chitin-triggered immunity [155,156]. Therefore, chitin-triggered immunity is an important immune pathway for host plants and is an extremely critical target for pathogens during pathogenesis.

#### 3.3.2. Antagonism with Anti-Microbial Compounds in Plants

Upon detecting infection by pathogens, plants secrete antimicrobial proteins/compounds that kill pathogen cells [157,158]. As a protective chemical barrier, they form a complex with microbial membrane steroids that compromise the plasma membrane integrity of the pathogen. Among them, saponins, such as oat saponins and α-tomatine, are common structural antimicrobial compounds necessary for the basic defense of plants against various pathogens.

Accordingly, certain effectors secreted by pathogens can act as detoxifying enzymes to degrade these antimicrobial compounds. For example, oat varieties lacking oat antimicrobials exhibit compromised resistance to the root-infesting fungi *Cryptosporidium graminis* and *F. graminearum* [157]. The fungus *Gaeumannomyces graminis*, which attacks oat roots, produces a saponin detoxifying enzyme that successfully infects oat tomatoes in which the steroidal glycol alkaloid α-tomatine is an antimicrobial compound that is resistant to fungal pathogens [157]. The leaf mold *C. fulvum* secretes a GH10 effector, CfTom1, whose function is required for α-tomatine detoxification [159]. In addition, strategies to resist attack by antimicrobial compounds include modified self-protection mechanisms, decorated on the surface of the mycelium, and the secretion of the virulence-promoting repeat effector Rsp3 by *U. maydis* [160]. Rsp3 is highly expressed during plant colonization and decorates the surface of fungal hyphae. Rsp3 binds to the maize antifungal proteins AFP1 and AFP2, which are similar in sequence to the antifungal mannose-binding protein Gnk2 secreted by Ginkgo. Overexpression of Rsp3 in maize barley increased resistance to AFP1, while silencing of AFP1 and AFP2 increased virulence [160], suggesting that Rsp3 protects fungal cells from the virulence effects of these antifungal proteins.

### 3.4. Interfering with Host Plant Cell Physiological Activities

#### 3.4.1. Regulation of Plant Gene Transcription

Effector proteins in the host plant nucleus function as transcription factors and reprogram the host defense pathway (Figure 4). The protein AvrBs3 of the pepper blotch bacterium *Xanthomonas campestris* pv. *vesicatoria* (Xcv), for example, has transcription activator-like (TAL) activity and acts as a transcription factor to directly induce the expression of plant genes [161,162]. AvrBs3 is an effector protein injected into plants by *X. campestris* pv. *campestris*, targeting UPA20, a major regulator of cell size. AvrBs3 causes hypertrophy of plant foliar cells by binding to the UPA20 promoter to facilitate pathogenic bacterial infestation and colonization [161]. A conserved target of the *X. campestris* pv. *campestris* TALE effector was recently found to be the AP2/ERF family transcription factor ERF121, and activation of the AP2/ERF transcription factor-dependent TALE promoted susceptibility to *X. campestris* pv. *campestris* through misregulation of plant defense pathways [163]. The transcription factors calmodulin-binding protein 60g (CBP60g) and SAR-deficient 1 (SARD1) in plants positively regulate SA-mediated signaling by directly binding to the promoters of SA synthesis-related genes [164,165]. *V. dahliae* secretes cysteine-rich small molecule proteins VdSCP7 and VdSCP41 to interact with CBP60g and SARD1 in plant nuclei to regulate immunity to fungal infection [122,165]. VdSCP41 binds to the transcriptional activation domain of CBP60g to repress its transcriptional activity [165]. The soybean rust fungus *Phakopsora pachyroot* effector candidate 23 (PpEC23) interacts with the soybean transcription factor GmSPL121 to suppress plant immunity. Unlike CBP60g and SARD1, GmSPL12l functions as a negative regulator of soybean defense [166]. The oomycete pathogen *Hyaloperonospora arabidopsidis* effector protein HaRxL21 interacts through the C-terminal EAR module to mimic the host plant recruitment of the transcriptional co-repressor Topless (TPL) to the transcriptional repressor site, suppressing plant immunity and enhancing host susceptibility to biotrophic and necrotrophic pathogens [167]. Both nuclear effectors, MoHTR1 and MoHTR2, secreted by the pathogenic bacterium *Inabaena ramorum*, translocate from the apoplast to the nucleus of initially penetrating and surrounding cells and reprogram the expression of immune-related genes by binding on effector binding elements in rice [168]. Interestingly, transgenic rice plants expressing these effectors exhibit increased susceptibility to hemibiotrophic pathogens but enhanced resistance to necrotrophic pathogens. RipAB, a type III effector secreted by the cyanobacterium *Ralstonia solanacearum*, was recently found to target the *Arabidopsis* TGACG sequence-specific binding transcription factor TGA. TGA directly activates the expression of RBOHD and RBOHF, whereas RipAB inhibits TGA expression by interfering with the recruitment of RNA polymerase II, repressing SA signaling, thus achieving successful infection [169]. Effectors also target post-transcriptional modifications; the wheat stripe rust *Puccinia striiformis* f. sp. *tritici* effector protein Pst_A23 regulates variable splicing of host disease-resistance-associated genes by binding to plant variable splice site-specific precursor RNA motifs to suppress host immune responses and promote pathogenicity [170].

#### 3.4.2. Degradation of Host Plant RNA

Ribonuclease-like effectors can interfere with host physiological activities by targeting RNA for degradation (Figure 4). *Blumeria graminis* secretes a group of effectors which are proteins with ribonuclease (RNase-like) folding (Ralphs). Transgenic expression of the ribonuclease-like effector CSEP0064/BEC1054 in wheat increases susceptibility to infection and inhibits ribosomal inactivation protein (RIP)-induced degradation of host ribosomal RNA, which helps to preserve living cells as a nutrient source for this biotrophic fungal pathogen [171]. Among them, BEC1054 may play a central role in this process, interacting with total RNA and producing virulence against wheat by targeting a variety of host proteins, including glutathione-S-transferase, malate dehydrogenase, the disease-course-associated proteins Pr5 and Pr10, and the elongation factors eEF1α and eEF1γ [168]. In addition, CSEP0064/BEC1054 physically interacts with PR10 in the nucleus and cytoplasm [171]. The effector Fg12, secreted by *F. graminearum*, also has ribonuclease activity, and Fg12 degrades total soybean RNA, induces plant cell death, and promotes pathogen virulence [172], similar to the ribonuclease VdRTX1 secreted by *V. dahliae*, which translocates to the plant cell nucleus causing cell death [173]. The enzymatically active ribonuclease effector Zt6 in the Septoria tritici blotch fungus *Zymoseptoria tritici* can also induce cell death, but Zt6 is nonessential for the virulence of the pathogen [174]. Two conserved ribonuclease effectors SRN1 and SRN2 are present in *Colletotrichum orbiculare*, in which SRN1 specifically cleaves the phosphate at the 3’ end of single-stranded RNA on guanosine, both of which are capable of cleaving RNA in the apoplast and enhancing the PTI response of the host [175].

#### 3.4.3. Interference with Plant Cell Degradation Pathways

Autophagy and the ubiquitin-proteasome system, as two of the most important intracellular degradation pathways, are very important for cellular homeostasis and the maintenance of normal cellular physiological functions (Figure 4). In recent years, there has been increasing evidence that the proteasome and autophagic pathways are central hubs of microbial effectors [176,177] and that the autophagy and protein-ubiquitin systems are essential for plant immunity to different pathogens [178] and have become common targets of many effectors.

Autophagy is a ubiquitous intracellular degradation process through which plants can resist external stresses and specifically degrade harmful substances that are detrimental to them and is essential for phytoplasma growth and differentiation. During early plant innate immunity, autophagy can be considered as a protective mechanism to limit pathogen-infection-induced programmed cell death (PCD). *Arabidopsis* autophagy-associated protein (ATG) plays an important role in autophagy [179]. PexRD54, an effector of the potato blight pathogen *Phytophthora infestans*, can bind the host autophagy protein ATG8CL to stimulate the formation of autophagic vesicles, deplete the autophagy receptor Joka2 in the ATG8CL complex, and interfere with the active role of Joka2 in pathogen defense [180]. The *P. syringae* effectors HrpZ1, HopF3 and AvrPtoB employ different molecular strategies to regulate autophagy. Calcium-dependent HrpZ1 oligomerization targets ATG4b-mediated ATG8 cleavage to enhance autophagy, while HopF3 also targets ATG8 but inhibits autophagy, both promoting infection through different mechanisms. AvrPtoB, on the other hand, enhances bacterial virulence by affecting ATG1 kinase phosphorylation and enhancing bacterial virulence [181]. Taken together, autophagy is likely to serve as a key target of pathogenic attack during infection, and, since autophagy is enhanced and inhibited by these effectors, this implies that autophagy may have different functions at different stages throughout the infection process.

The *M. oryzae* effector AvrPiz-t, on the other hand, targets the host ubiquitin-proteasome system to manipulate plant defense. AvrPiz-t is able to bind to the E3 ubiquitin ligases APIP6 and APIP10 [182,183] to mount an attack on the host ubiquitin-proteasome system, and APIP10 promotes the degradation of Piz-t through the 26S proteasome system. *P. syringae* HopZ4, a member of the YopJ family T3E, interacts with Rpt6, an AAA ATPase subunit of the 26S proteasome 19S RP, and inhibits proteasome activity during infection [184]. The *U. maydis* effector Tin2 masks the ubiquitin-proteasome degradation motif in ZmTTK1 in maize, thereby stabilizing the active kinase and activating genes involved in anthocyanin biosynthesis [185]. Interestingly, Tin2 homologs in *U. maydis* and the related black cob pathogen *Sporisorium reilianum* function differently, targeting different ZmTTK analogs, leading to stabilization and inhibition of protein kinases, respectively [186], whereas only *U. maydis* Tin2 induces anthocyanin biosynthesis, suggesting that Tin2 in *U. maydis* may be newly functionalized to promote a pathogenic lifestyle [186]. In wheat, the SNF1-related kinase TaSnRK1α interacts with an orphan protein, TaFROG, whose expression is induced by the fungal toxin deoxynivalenol (DON). Overexpression of TaFROG stabilizes TaSnRK1α and enhances resistance to *Fusarium oxysporum* in wheat [187,188]. The cytoplasmic effector Osp24 in *F. graminearum* promotes the degradation of TaSnRK1α by competing with TaFROG for binding to the ubiquitin-26S proteasome [188]. The Septoria tritici blotch pathogen *Z. tritici* secreted protein ZtSSP2 interacts with wheat E3 ubiquitin ligase (TaE3UBQ) in yeast, and downregulation of TaE3UBQ increases the susceptibility of wheat to STB. This implies that ZtSSP2 likely promotes infestation by inhibiting the role of TaE3UBQ in immunity [189].

#### 3.4.4. Interference with Host Plant Protein Function

There are also some effectors that target functional proteins in the host plant cytosol, causing functional abnormalities and thus pathogenic effects, often through alterations in protein localization (Figure 4). A typical example is the *P. syringae* effector HopI1, which hijacks the protein target Hsp70 to the chloroplast, resulting in altered thylakoid structure and inhibition of SA accumulation [190,191]. HopNI is another well-studied *P. syringae* effector protein encoding a cysteine protease that cleaves an intrinsic protein of photosystem II in tomato cells PsbQ, thereby reducing water photolysis. The haustorium-specific effector protein Pst_12806 in the wheat strain *Puccinia striiformis* f. sp. *tritici* then translocates into the chloroplast and affects chloroplast function. Pst_12806 interacts with the C-terminal Rieske domain of the wheat TaISP protein (a putative component of the cytochrome b6-f complex) and interferes with TaISP function in the electron transport chain, leading to a reduction in the rate of plant electron transport and diminished ROS accumulation, which in turn inhibits the expression of defense-related genes [192]. In addition, the nucleophile integrin-like effector SsITL interacts with the calcium-sensitive receptor (CAS) in chloroplasts and interferes with CAS-associated SA-mediated immunity [193,194]. The soybean pathogenic fungus *Phytophthora sojae* uses an important effector PsAvh262 to stabilize the endoplasmic reticulum (ER)-lumen-bound immunoglobulin (BIPS), a negative regulator of plant resistance to blast fungus. By stabilizing BIPS, PsAvh262 inhibits ER stress-induced cell death and promotes eubacterium infection. The direct targeting of ER stress regulators may represent a common mechanism for microbial manipulation of the host [195]. The *M. oryzae* effector MoCDIP4 targets the mitochondria-associated OsDjA9-OsDRP1E protein complex to promote the degradation of OsDRP1E, which plays a role in mitochondrial division. The DnaJ protein OsDjA9 interacts with the kinesin-associated protein OsDRP1E, while MoCDIP4 competitively binds OsDjA9, causing OsDRP1E accumulation, resulting in abnormal mitochondrial development [196] and reduced immunity in rice. The *M. oryzae* effector AvrPiz-t interferes with the binding of OsAKT1, a K^+^ channel protein localized on the rice plasma membrane, to the upstream regulator cytoplasmic kinase OsCIPK23, inhibiting OsAKT1-mediated K^+^ signaling to regulate host immunity [197]. The soil-borne fungus *V. dahliae* effector PevD1 indirectly activates CRY2 through antagonism with an asparagine-rich protein, NRP. In cotton and *Arabidopsis*, NRP interacts with cryptoglobin 2 (CRY2), which functions in the nucleus [198,199], and NRP, in a blue-light-independent manner, tethers CRY2 in the cytoplasm, resulting in CRY2 dysfunction [200], thereby delaying the induction of early flowering in cotton and *Arabidopsis* by *V. dahliae*. The potato late blight pathogen *Phytophthora infestans* RXLR effector Pi04314 interacts with three host protein phosphatase 1 catalytic (PP1c) isoforms, leading to their relocation from the nucleolus to the nucleoplasm to promote colonization of blast-infested leaves and attenuate the induction of JA and SA-responsive genes [201]. The wheat stripe rust effector PstGSRE1 interacts with the ROS-associated zinc-finger transcription factor TaLOL2 [202,203]. This interaction prevents the nuclear localization of TaLOL2, thereby impairing plant defenses by inhibiting H_2_O_2_ accumulation and ROS-mediated cell death.

#### 3.4.5. Interference with Host Vesicle Transport

In the game played between plants and pathogens, vesicular transport plays an important role in both plant hosts and pathogenic microorganisms (Figure 4). Recent studies in the maize black cob disease pathogen *Ustilago maydis* suggest that extracellular vesicles (EVs) may be involved in host pathogenesis [204]. A number of mRNAs for known effectors and other virulence-related proteins were found in EVs, some of which did not contain the predicted secretory signal. This suggests that EVs are likely to act as a delivery mechanism, transporting different RNAs for intercellular communication between pathogens and the host.

The guanine nucleotide exchange factor (GEF), an ADP-ribosylation factor, is essential for vesicle trafficking. The *P. syringae* type III effector HopM1 interacts with the host ADP ribosylation factor-guanine nucleotide exchange factor AtMIN7, which plays an important role in both PTI and ETI processes. HopM1 induces AtMIN7 degradation in a proteasome-dependent manner, which in turn promotes pathogen infection [205,206]. The *M. oryzae* zinc finger transcription factor MoCRZ1 regulates vesicle-mediated secretion-related genes, suggesting that MoCRZ1 may be involved in effector secretion [207]. The *B. graminis* f. sp. BEC4 functions as a virulence factor by blocking defense-related vesicle transport [208]. A potential host target for BEC4 is the ATP ribosylation factor-GTPase activating protein (ARF-GAP), which is a key regulator of eukaryotic membrane transport [208]. *Phytophthora brassicae* secretes the highly conserved effector RxLR24, which interacts with the Raba GTPase family member RABA1a in vesicles and plasma membranes, inhibiting the function of Raba GTPase in vesicle secretion and reducing vesicle secretion of the antimicrobial agents PR-1, PDF1.2, and other potentially defense-related compounds [209]. The RXLR-WY effector PexRD31 in *Phytophthora infestans* increased the number of FYVE-positive vesicles in *N. benthamiana* cells [210]. FYVE-positive vesicles also accumulated in leaf cells near *P. infestans* mycelium, suggesting that the pathogen may enhance endosomal transport during infection.

### 3.5. Manipulating Plant Downstream Immune Responses

#### 3.5.1. Interference with Plant Hormone Signaling

To protect themselves from the attack of pathogens, plants rely on the fine regulatory network of phytohormones to resist disease. On the other hand, pathogens act through effectors to evade or disrupt phytohormone-dependent plant defenses [211,212,213,214,215]. The pathogen lifestyle determines the type of phytohormone signaling intervention. These effectors mainly achieve pathogenicity by inhibiting or circumventing some of the host phytohormone biosynthesis and/or signaling pathways involved in pathogen resistance [213]. In addition, some pathogens can produce phytohormones and their derivatives that mimic the host to manipulate or hijack host hormone homeostasis [20].

The role of SA, JA and ethylene (ET) in plant disease resistance has been extensively studied, with the SA pathway being activated mainly when invaded by biotrophic pathogens [216] and JA being triggered mainly against necrotrophic pathogens, with signaling crosstalk between both pathways usually manifesting mutual antagonism [217]. Recent results have expanded this network of crosstalk among hormones include ET, ABA, and BR, suggesting a more systemic mode of action for phytohormone-mediated immunity, with strong evidence that all these hormonal pathways are targets of effectors [215].

SA can be transformed in the chloroplast from chorismate, the final product of the shikimate pathway [218,219,220,221,222], and is involved in systemic acquired resistance (SAR) and induced systemic resistance (ISR) by hormone signaling [223]. SAR is a broad-spectrum type of resistance induced in host plants after primary infection, which can protect uninfected parts of plants from subsequent infections [54,224]. Like SAR, ISR is also a systemic resistance process, which depends on hormone-induced signal transduction, such as ET and JA, and is accompanied by the activation of pathogenesis-related genes [225]. There are two main ways that pathogens interfere with SA, direct and indirect [226,227]. On the one hand, the effector acts directly on the SA pathway, including biosynthesis and degradation (Figure 5). For example, *U. Maydis* secretes the chorismate mutase Cmu1 and imports it into the phenylpropane pathway. Chorismate mutase is a key enzyme in the shikimate pathway, and can catalyze the conversion of chorismate into prephenate, the precursor of tyrosine and phenylalanine, to inhibit SA biosynthesis [211]. The oomycete pathogens *P. sojae* and the fungus *V. dahliae* secrete isochorismatases PsIsc1 and VdIsc1, respectively [16], and they can convert isochorismate into 2,3-dihydro-2,3-dihydroxybenzoate (DDHB) and pyruvate, making isochorismate unavailable for SA biosynthesis [215]. In addition, effectors can degrade SA, such as *Ralstonia solanecearum* POP [228] encoding a type III secreted effector in the AvrE family, inhibiting the SA-mediated defense in tomato roots and stems, which are the natural infection sites by *R. solanacearum*. Some effectors of *Sclerotinia sclerotiorum* have also been found to have the ability to degrade SA [229]. On the other hand, effectors can act indirectly on the regulation of SA; the effector non-expressor of pathogenesis-related genes 1 (NPR1) is a key regulator of the SA pathway [230,231], related to cold, salt and oxidative stress tolerance and plant immunity [232], and is required for activation of PR gene expression. The conserved effector PNPi (Puccinia NPR1 interactor) of stripe rust *Puccinia striiformis* f.sp. competes with the transcription factor TGA2.2 for binding to wheat NPR1. Therefore, PNPi may reduce PR gene expression by interfering with the interaction between NPR1 and TGA transcription factors [233]. The type III effector AvrPtoB of *P. syringae* also targets NPR1 and inhibits NPR1-dependent SA signaling through the host 26S proteasome in a manner dependent on the E3 ligase activity of AvrPtoB [234]. The fungal effector RxLR48 promotes nuclear localization of NPR1 and inhibits its proteasome-mediated degradation, suppressing SA signaling to subvert plant innate immunity [235].

JA can function under SA-dependent or independent conditions and modulate plant immunity to infection by hemibiotrophic pathogens [236,237], thus also becoming an important target for effector attack. For example, AvrB, a type III effector of *P. syringae*, interacts in plant cells with MPK4, HSP90, and RIN4 to interfere with hormone signaling, cause upregulated expression of JA-responsive genes, and enhance plant susceptibility [63]. Another *P. syringae* effector, HopZ1a, directly interacts with and degrades the JA repressor JAZ and activates JA signaling [238]. So do the effectors HopBB1 [239] and HopX1 [64], the latter promoting their degradation in a COI1-independent manner. Some pathotypes of *Fusarium oxysporum* can produce bioactive isoleucine and leucine-conjugated JA (JA-Ile/Leu) as virulence effectors to promote root infection or aboveground infection [240], and their virulence-promoting effector Fo5176-SIX4 may activate JA signaling for successful infection [241]. The necrotrophic pathogen *Lasiodiplodia mediterranea* in grapes produces the JA ester lasiojasmonate A (LASA) as an inactive JA reservoir that can be converted into the bioactive form JA-Ile. Although JA often activates plant defenses against necrotrophic pathogens, LASA is thought to be a metabolite effector that induces JA-mediated cell death at late infection stages, leading to successful necrotrophic colonization [242]. The *U. maydis* effector JA/ET signaling inducible factor 1 (Jsi1) was recently found to interact with several members of the TPL/TPR protein family of plant co-repressors [243]. Jsi1 interacts with the WD40 domain of TPL/TPR, resulting in downregulation of the ERF branch of JA/ET signaling. Jsi1 in *Arabidopsis* and maize expression activates the JA/ET pathway and leads to susceptibility to biotrophic pathogens [243]. In contrast to the above-mentioned activation of the JA pathway, *M. oryzae* deploys an antibiotic biosynthetic monooxygenase effector, ABM, for the conversion of fungal- and host-derived JA to hydroxylated JA, a fungal 12OH-JA secreted during host penetration that helps evade defense responses [212].

ET is also a classical plant defense hormone, whereby the activation of ET signaling makes plants resistant to pathogenic bacterial attack [244,245,246,247]. ET is produced by S-adenosylmethionine (S-ADOMet) catalyzed by ACC synthase (ACS), and ET production is directly related to ACS activity [248,249,250,251]. *P. syringae* uses HopAF1 to target the methionine cycle that disrupts ET biosynthesis [252], whereas the soybean blast RxLR polymorphic effector PsAvh238 promotes soybean blast infection by destabilizing GmACSs and inhibiting ET biosynthesis [253]. In contrast, the necrotrophic fungus *Cochliobolus miyabeanus* requires activation of ET signaling to cause rice brown spot disease [254]; blocking the mechanism of hijacking the ET signal significantly reduced the colonization of *C. miyabeanus* on rice leaves, suggesting that *C. miyabeanus* is likely to use ET as a metabolite effector to promote virulence.

Recently, growth-related phytohormones, such as auxin, cytokinin (CKs), brassinosteroid (BR), abscisic acid (ABA), and gibberellin (GA) have also been shown to modulate immune defense in plants [255,256,257,258]. For example, the RxLR effector penetration-specific effector 1 (PSE1) of the oomycete *Phytophthora parasitica* infecting *Arabidopsis* is transiently upregulated during host root infection [259], and overexpression of PSE1 can reduce the accumulation of auxin and increase susceptibility to *P. parasitica*. The cysteine protease encoded by *P. syringae* type III effector AvrRpt2 activates the auxin pathway in *Arabidopsis* by promoting the proteasome-dependent destabilization of AUX/IAA, and pathogen susceptibility [260]. *Arabidopsis* plants expressing AvrRpt2 exhibit increased sensitivity to exogenous auxin and increased levels of endogenous free auxin [261]. Another type III effector in *P. syringae*, HopQ1, induces CK signaling in *Arabidopsis* and inhibits FLS2 accumulation, thereby suppressing defense signaling and promoting disease progression [262]. The effector protein EqCSEP01276 of the rubber powdery mildew fungus *Erysiphe quercicola* inhibits ABA biosynthesis by interfering with the localization of the key ABA biosynthesis enzyme 9-cis-epoxycarotenoid dioxygenase 5 (HbNCED5) in the chloroplast, thereby inhibiting the host defense [263].

Furthermore, due to the extensive interactions among plant hormone signaling pathways, this often leads to synergistic or antagonistic functions [258,264,265,266,267,268,269], such as antagonism between SA and JA pathways [270,271,272], and synergism between JA and ET pathways [273,274,275]. Thus, some pathogens are also able to use phytohormone crosstalk to promote disease development, i.e., the pathogen mediates the activation of a phytohormone signaling pathway by inhibiting another phytohormone pathway that leads to resistance [276,277]. For example, the effector SnTox3 not only stimulates ethylene biosynthesis and signaling pathways, but also reduces the active form of CK by oxidative degradation in both ET-dependent and ethylene-independent ways. SnTox3-mediated activation of the ET signaling pathway reduces active CKs and inhibits SA signaling pathways and ROS production, thereby promoting susceptibility to pathogens in wheat [278].

#### 3.5.2. Utilization of RNA Silencing Strategy

A recent study in apple showed that sRNAs in plants indirectly regulate R gene expression by targeting genes associated with R gene co-expression, thereby contributing to a negative feedback loop [279], suggesting that the role of sRNAs in ETI is likely to be more important than was thought. Silencing suppression based on small RNAs is a common strategy used by different types of plant fungal pathogens to promote infection [19,280,281,282]. The secreted effector protein PgtSR1 encoded by two alleles of the wheat stem rust *Puccinia graminis* f. sp. *tritici* inhibits RNA silencing in plants and hinders plant defense by altering the abundance of small RNAs that act as defense regulators. This effector partially suppresses R protein-induced cell death and enhances resistance to disease susceptibility to a variety of pathogens [283]. The necrotrophic fungus *Sclerotinia sclerotiorum* produces a variety of different high abundance sRNAs during infection, while sRNAs in host plants are significantly downregulated compared to pre-infection, suggesting that *S. sclerotiorum* sRNAs may contribute to the silencing of immune components in plants [284]. RNA silencing repressors PsPSR1 and PsPSR2 in the oomycete pathogen *P. sojae* [285,286,287,288] inhibit RNA silencing in plants by specifically suppressing secondary siRNA biogenesis thereby promoting infection, and the ectopic expression of these RNA silencing repressors enhances the susceptibility of plants to viruses and blast molds. This suggests that some eukaryotic pathogens have evolved virulence proteins that target the host RNA silencing process to promote infection.

In addition to infestation strategies targeting host sRNAs, the pathogen itself produces sRNAs as effectors [19]. *B. cinerea* produces sRNAs through the action of the dicer-like proteins BC-DCL1 and BC-DCL2, and some small RNAs are then transported into *Arabidopsis* cells to hijack the host RNA interference (RNAi) machinery. sRNA effectors bind to *Arabidopsis* Argonaute1 (Ago1) and selectively repress the expression of host immune-related genes. Naturally occurring cross-kingdom RNAi represents an advanced virulence mechanism [19]. A novel miRNA-like RNA, PST-milR1, was efficiently expressed in *P. striiformis* f. sp. *tritici.* pST-milR1 is translocated into plant cells and represses the expression of the wheat PR2 gene. Silencing of the PST-milR1 precursor gene results in the virulence of *P. striiformis* f. sp. *tritici* on wheat CYR31 isolates being diminished, suggesting that PST-milR1 is a pathogen effector that suppresses natural immunity in wheat [281].

#### 3.5.3. Regulation of Reactive Oxygen Species Generation

Pathogen-induced ROS are hallmark cell signaling molecules that initiate plant immune responses [289,290]. ROS directly damage DNA, RNA, polysaccharides, lipids, proteins, and smaller metabolites, and can also initiate various defense responses, such as phytoalexin biosynthesis and activation of defense-related genes [291]. Outside the cell, apoplast ROS are mainly produced by peroxidases; NADPH oxidases, commonly referred to as respiratory burst oxidase homologs (RBOHs), act as ROS at the plasma membrane after peroxidase-initiated oxidative bursts amplifiers come into play [292]. Intracellularly, ROS are produced by different organelles, including chloroplasts, mitochondria, peroxisomes, and the endoplasmic reticulum in plants [293,294,295].

For successful colonization, pathogens have developed different strategies to neutralize or inhibit ROS production (Figure 5). The rice blast fungus secretes the peroxidase-peroxidase CPXB to prevent the accumulation of ROS in rice epidermal cells during early infection. The maize peroxidase gene POX12 is involved in the production of ROS, and the *U. maydis* effector PEP1 directly targets POX12 by scavenging ROS in the apoplast, thereby inhibiting the ability to produce ROS and early defense responses [131,203,296]. The extracellular superoxide dismutase PsSOD1 of the wheat stripe rust *P. striiformis* f. sp. contributes to successful infection by scavenging host-derived ROS [297]. In the cytosol, NADP-malic enzyme (NADP-MES) catalyzes the oxidative decarboxylation of L-malate to produce pyruvate, CO_2_, and NADPH, which is the electron (e^−^) donor for ROS production by NADPH oxidase [298]. The *B. cinerea* nontoxic protein AVR-Pii interacts with OsNADP-ME2, which is essential for ROS accumulation in rice [299,300], and specifically inhibits ROS burst and NADP-ME activity in vitro [299]. In *B. cinerea*, an iron-binding SSPs family effector BcIBP prevents *Arabidopsis* ROS formation by limiting intracellular metal accumulation [301]. In *Arabidopsis*, ROS-mediated defense is controlled by BPA1 (the binding partner of ACD11, whose deletion leads to accelerated cell death) and its homologs, such as BPLs and BPA1-like proteins. RxLR207 of *Phytophthora capsici* promotes the degradation of BPA1, BPL1, BPL2, and BPL4 in a 26S proteasome-dependent destabilization of ACD11 to promote the accumulation of ROS and activation of PCD to further exploit host resources [235]. Phosphorylation of NAD^+^ by the *Xanthomonas* effector AvrRxo1 can reduce ROS bursts in tobacco immune response [302]. *P. striiformis* f. sp. effector Pst18363 has been shown to target and stabilize the negative defense regulator TaNUDX23 (Nudex hydrolase), which inhibits ROS accumulation to promote pathogen infection [303]. The *F. oxysporum* apoplasts (SIX1 and Foa1) and cytoplasmic effectors (Avr2, Foa2 and Foa3) promote host colonization by inhibiting Flg22- or chitin-induced ROS [304]. Recent studies have shown that AVR-Pita, an effector of *M. oryzae*, interacts with the cytochrome c oxidase (COX) assembly protein OsCOX11, a key regulator of reactive oxygen metabolism in rice mitochondria [305]. AVR-Pita enhances COX activity in mitochondria, thereby inhibiting ROS accumulation.

#### 3.5.4. Manipulation of Plant Cell Death

In plant-pathogen interactions, recognition of pathogens by NLR and PRR usually leads to HR [306], and effectors achieve infection by manipulating plant cell death. Programmed death confines pathogens to a small number of infected cells, while suppression of HR promotes pathogen infection.

Triggering the HR response is a common strategy for necrotrophic pathogens, and transient expression of the rice fungus effectors (MoCDIP1 to MoCDIP5) induced cell death in rice protoplasts and *N. benthamiana*, suggesting that they function in the necrotrophic phase. The *B. cinerea* BcCrh1 protein acts as a cytoplasmic effector and inducer of plant defense, catalyzing the cross-linking of titin and glucan polymers in the fungal cell wall. BcCrh1 is localized in the vesicles and endoplasmic reticulum during saprophytic growth. However, after plant infection, the protein accumulates in the infection cushion and is then secreted into the apoplast and translocated into plant cells where it induces cell death and defense responses [307]. In contrast, some pathogens promote their own infestation by inhibiting the HR response, such as the *P. syringae* type III effector HopS2, which shows extremely strong HR inhibitory activity [308]. Overexpression of the wheat stripe rust effector PstCFEM1 suppresses cell death, ROS accumulation and callose deposition triggered by Pst322, an elicitor-like protein of PST [309].

In addition, pathogens secrete another group of protein-like secondary metabolites called host-selective toxins (HSTs). HSTs enhance PCD by promoting ROS accumulation through activation of host Nox or MAPK signaling involving Ca^2+^ influx [203]. The host-selective toxins ToxA and ToxB secreted by the necrotrophic pathogen *P. tritici-repentis* [18] is a typical example, where PtrToxA interacts with ToxABP1 in wheat chloroplasts. ToxABP1 silencing enhances intracellular expression of PtrToxA in different plants, decreases the level of photosystem I and II, and leads to necrotic phenotypes to promote the proliferation of necrotrophic fungus [310,311]. SnToxs, including SnToxA, SnTox1, and SnTox3, in the powdery mildew fungus *Stagospora nodorum* Berk. have similar roles to cause necrosis [312,313,314].

Many CWDEs induce cell death in an enzyme activity-dependent or -independent manner. *F. oxysporum* FoEG1 (GH12), *B. cinerea* xylanase BcXYG1 (GH12), and xylanase BcXY1 all induce plant cell death in an enzyme activity-independent manner as PAMPs [315,316,317], while the β-1,3-glucanases CfGH17-1 and CfGH17-5 of biotrophic pathogens induce cell death by degrading the released cell wall products [79]. The xylanase RcXYN1 of *S. graminis* also triggers cell death ROS production in wheat and promotes its infection [318].

Certain conserved domains are also capable of triggering cell death in plants. The ability of effectors containing necrosis- and ethylene-inducing peptide domains (NEPs) to induce plant cell death has been demonstrated, such as the NEP1-like protein (NLP) of oomycete origin in Arabidopsis, which triggered light-dependent cell death, as well as post-translational activation of mitogen-activated protein kinase activity, callose deposition, nitric oxide, and reactive oxygen intermediates [319]. Similarly, BeNEP1 and BeNEP2 were identified in the lily pathogen *Botrytis elliptica* [320], SsNEP1 and SsNEP2 in *S. sclerotiorum* [321], and NLP1 and NLP2 in *V. dahlia* [322]. Interestingly, NLP is also widely present on other trophic types of hosts, not limited to necrotrophic pathogens. In the hemibiotrophic fungus *Colletotrichum higginsianum*, ChNLP1 and ChNLP2 [323] are specifically expressed during the transition to necrotrophy in tobacco and induce effective cell death when overexpressed. NLP proteins were also found in the obligate parasite downy mildew pathogen *Hyaloperonospora arabidopsidis*, but none of them had the ability to induce cell death and appeared to be non-virulent [324].

## 4. Conclusions and Perspectives

Effectors are an essential part of plant-pathogen interactions—they exert pathogenic effects mainly by targeting R proteins in the plant. We have classified and summarized their mechanisms of action at different stages of infestation and found that effectors cover a wide range of secreted substances, and a large number of effectors exist at all stages of pathogenicity, reflecting the involvement of plants and pathogens in a high-speed evolutionary arms race. Although current research on effectors is quite prolific, and many effectors with clear mechanisms of action have been revealed, as the iceberg model proposed by Thordal-Christensen [35] suggests, there are more effectors, yet undiscovered, hiding under the iceberg in the large number of crosstalk signals between the interacting units. Based on existing research progress, this taxon of effectors is extremely broad in scope and exhibits surprising diversity [29]. Therefore, to discover commonalities among these molecular sequence diversities will be a major challenge. In this regard, some explorations of effector lineages and their functional characterization have already been carried out by Kanja [325], Jaswal [326], and others. Interestingly, the arms race between plants and pathogens provides insights into the fact that, to keep effectors at the forefront of evolution, they are usually located in duplication-rich and gene deficient regions on chromosomes [327]. This makes them more likely to have an unparalleled rate of evolution. Intriguingly, the emergence of new technologies has brought with it new developments, through the combination of genomics, transcriptomics, metabolomics, and proteomics, as well as live-cell imaging techniques which rely on new molecular probes and sensors, which will help us to understand the full picture of pathogen genomes more efficiently and systematically and to mine effector genes.

On the other hand, effectors often exhibit a multifunctional character [328], with some effectors targeting multiple proteins involved in immunity to achieve different virulence/non-virulence functions [30], and others moving between cellular components to generate a chain of responses [329]. This undoubtedly challenges us to clarify the specific pathological models of effector regulation in plant defense. In this complex signaling network, important questions include: Which physiological pathways of the plant do pathogenic bacteria focus on manipulating? Is there also a signaling link between effectors that work together? How do they achieve targeted transport and delivery in plant cells? We look forward to exciting results from studies and answers to these questions, as well as to the holistic understanding of effector mechanisms and functions that will be achieved.

## Figures and Tables

**Figure 1 ijms-23-06758-f001:**
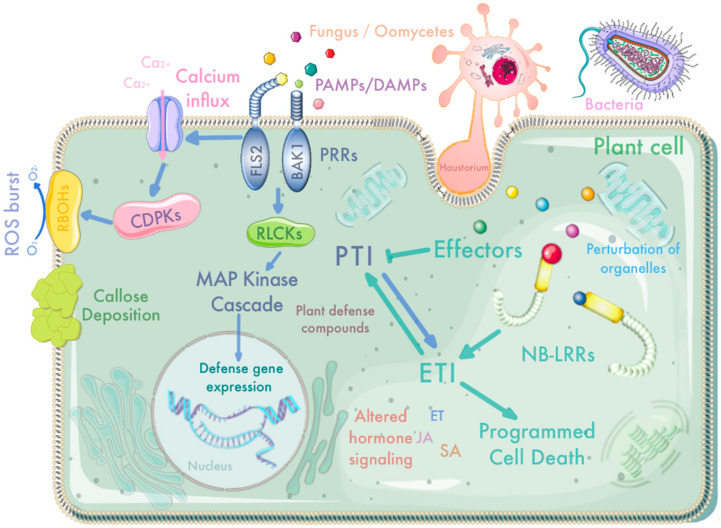
Plant immune system. PRRs of plant cells recognize PAMPs and trigger the first layer of plant immune response PTI, which triggers responses such as calcium inward flow and reactive oxygen species burst. Pathogens produce effectors to inhibit PTI, and plant R proteins, such as NB-LRR proteins, are activated by effectors to produce the second layer of immune response ETI, which disrupts effectors and causes cell death.

**Figure 2 ijms-23-06758-f002:**
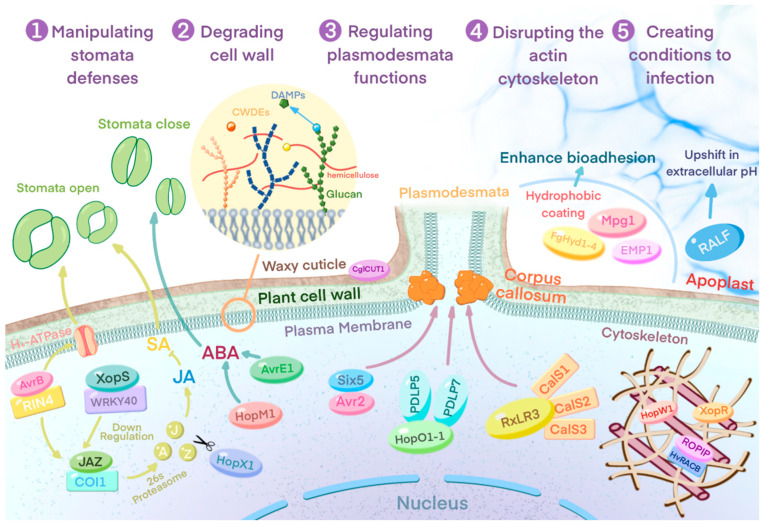
Schematic diagram of effectors that break physical barriers to infestation and regulate the infestation environment. Effectors promote infestation by regulating stomata, degrading cell walls, regulating intercellular filament function, disrupting the cytoskeleton, and creating conditions conducive to infestation. *P. syringae* AvrB interacts with RIN4 protein to promote COI1 and JAZ protein binding and enhance plasma membrane H^+^-ATPase AHA1 activity. HopX1 inhibits stomatal immunity by degrading multiple JAZ transcriptional repressors leading to JA activation. xopS stabilizes WRKY40 and decreases JAZ expression. HopM1 and AvrE1 target the ABA signaling pathway, increase ABA accumulation, and induce stomatal closure. *Fusarium oxysporum* Avr2 and Six5 interact with intercellular filaments to enlarge the pore size. The oomycete pathogen *Phytophthora brassicae* RxLR3 effector physically interacts with callus synthases CalS1, CalS2 and CalS3 to impede callus accumulation in intercellular hyphae. *P. syringae* HopO1-1 interacts with and destabilizes the PD proteins PDLP7 and PDLP5. HopW1 disrupts the actin cytoskeleton by forming complexes with actin. *Bradyrhizobium graminearum* ROPIP targets the barley ROP GTPase HvRACB and manipulates host cell microtubule organization to facilitate its own cell entry. *Xanthomonas oleifera* T3E XopR binds to actin in the cell cortex to manipulate actin assembly and disrupt the host actin cytoskeleton. The hydrophobic protein MPG1 in *Fusarium inermis*, FgHyd1~FgHyd4 in *F. graminearum*, and the extracellular matrix protein EMP1 in *F. graminearum* can create hydrophobic spaces and participate in appressorium development. Ralf-like effectors can cause an increase in extracellular pH to promote the invasive growth of the fungus. Overall, effectors favor their own infestation by modifying the structural properties of plant cells and environmental conditions.

**Figure 3 ijms-23-06758-f003:**
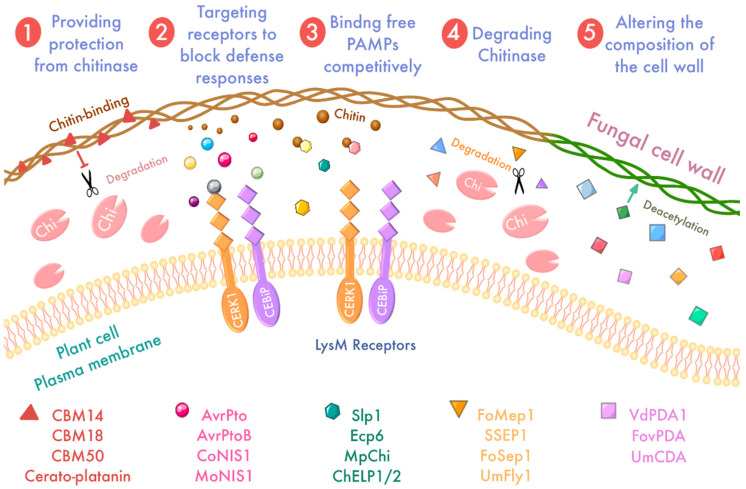
Inhibition of PAMP-triggered immunity by pathogens. Pathogens have developed different strategies to suppress PAMP-triggered immunity. For chitin, the main means used by pathogens are: (i) protection of the mycelium from degradation by plant chitinases, with representative effectors being the CBM family and CPP proteins; (ii) inhibition of LysM receptor recognition, with representative effectors being AvrPto, AvrPtoB, CoNIS1 and MoNIS1; (iii) isolation and masking of chitin oligosaccharides, representative effectors are Slp6, Ecp6, MpChi and ChELP1/2; (iv) targeting chitinases for degradation, representative effectors are FoMep1, SSEP1, FoSep1 and Umfly; and (v) modification and transformation of cell wall components, representative effectors are VdPDA1, FovPDA, UmCDA.

**Figure 4 ijms-23-06758-f004:**
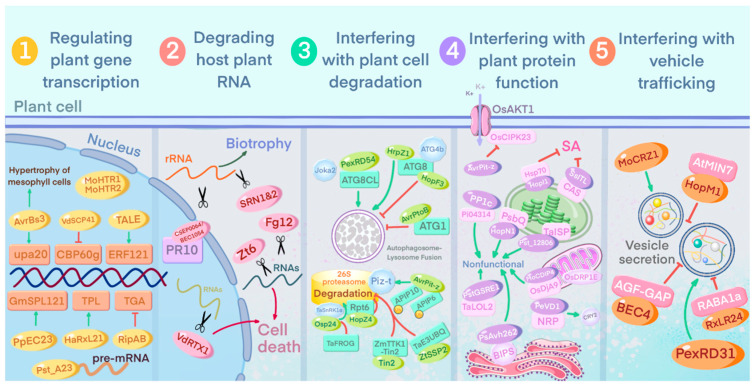
Effectors interfere with host plant cell physiological activities. AvrBs3, VdSCP41, TALE, PpEC23, HaRxL21, and RipAB have transcriptional activator-like activity and act as transcription factors that interfere with host cell degradation pathways, host protein function, and host vesicle transport, through regulating transcription of different genes, thus directly inducing the expression of plant genes. Nuclear effectors of *M. oryzae* MoHTR1 and MoHTR2 also reprogram the expression of immune-related genes in rice. In addition, the wheat stripe rust effector protein Pst_A23 suppresses host immune responses by regulating variable splicing of host disease-resistance-associated genes. *Fusarium graminearum* CSEP0064/BEC1054, *F. graminearum* Fg12, the secreted ribonuclease VdRTX1 of *V. dahliae*, *C. orbiculare* SRN1 and SRN2, and the wheat leaf blight effector Zt6 degrade host RNA and cause cell death. *Phytophthora infestans* effector PexRD54, *P. syringae* bacterial effectors HrpZ1, HopF3, and AvrPtoB employ different molecular strategies to regulate autophagy. *P. syringae* HopZ4, avrPiz-t, the *U. maydis* effector Tin2 launch an attack on the host ubiquitin-proteasome system to inhibit proteasome activity. Osp24 in *F. graminearum*, *ZymosepVictoria tritici* effector ZtSSP2, *P. syringae* effector HopI1, HopN1, wheat stripe rust Pst_12806, PstGSRE1, nucleophile integrin-like effector SsITL, *Phytophthora sojae* effector PsAvh262, AvrPiz-t, *V. dahliae* effector PevD1 and *Phytophthora infestans* effector Pi04314 act on different protein targets to exert pathogenic effects. *P. syringae* effector HopM1, rice blast fungus zinc finger transcription factor MoCRZ1, *Blumeria graminis* BEC4, *Phytophthora brassicae* RxLR24 effector and RXLR-WY effector PexRD31 in *Phytophthora infestans* are involved in the regulation of vesicle secretion.

**Figure 5 ijms-23-06758-f005:**
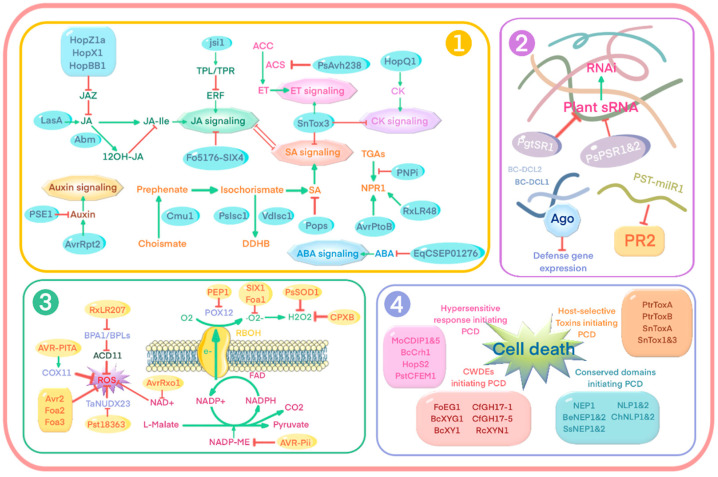
Effectors manipulate downstream immune responses in host plants. Effectors mainly manipulate plant downstream immune responses by interfering with plant hormone transduction, disrupting plant RNA silencing, regulating ROS generation, and manipulating plant cell death. The effector chorismate mutase Cmu1 of *Ustilago maydis*, the oomycete pathogen *Phytophthora sojae* and the fungus *Verticillium dahliae* effectors PsIsc1 and VdIsc1, POPs in *Ralstonia solanecearum*, *P. striiformis* f.sp. effector PNPi, *P. syringae* effector AvrPtoB, HopZ1a, HopBB1, HopX1, HopAF1, AvrRpt2, HopQ1, *Phytophthora capsicum* effector RxLR48, *F. oxysporum* Fo5176-SIX4, rice blast effector ABM, LASA of *L. medterranea, Phytophthora sojae* RxLR effector PsAvh238, Phytophthora effector PSE1 and EqCSEP01276 of *Erysiphe quercicola* SnTox3, interfere with the transduction of different hormone signals in plants. The wheat stem rust *Puccinia graminis* f. sp. *tritici* effector protein PgtSR1, PsPSR1 and PsPSR2 in *Phytophthora sojae*, *Botrytis cinerea* BC-DCL1 and BC-DCL2 and PST-milR1 are gene silencing mechanisms. The maize smut effector PEP1, the biotrophic pathogen *Puccinia striiformis* PsSOD1 and avirulent proteins AVR-PII, AVR-PITA in *M. oryzae*, inhibit ROS accumulation. The *M. oryzae* effector enhances COX activity by interacting with mitochondrial OsCOX11; OsCOX11 is a key regulator of mitochondrial ROS metabolism in rice. RxLR207 in *Phytophthora capsici* can promote the degradation of BPA1, BPL1, BPL2, and BPL4, and destabilize ACD11 in a 26S proteasome-dependent manner to promote ROS accumulation and PCD activation. *F. oxysporum* apoplasts (SIX1 and Foa1) and cytoplasmic effectors (Avr2, Foa2 and Foa3) promote host colonization by inhibiting flg22-induced or chitin-induced ROS. There are four main types of effectors that cause cell death, namely effectors that trigger HR responses, host-selective toxins, cell wall-degrading enzyme effectors, and effectors with conserved necrosis-inducing domains.

## Data Availability

Not applicable.

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
