# Peer review of "Action Mechanisms of Effectors in Plant-Pathogen Interaction"

_ijms, 2022, doi:10.3390/ijms23126758_

Round 1

Reviewer 1 Report

The present manuscript is a useful review of literature related to the effector mediated plant-pathogen interaction that underlies plant resistance and susceptibility to diseases.

Overall, the review is well-structured and easy to read, making it surely useful to readers that approach the subject of plant-pathogen interaction.

There are however some suggestions I would make to the authors:

1) I do not find the title to be effective and reflecting faithfully the content of the review. While effectors are often essential in the establishment of a disease, as the authors do show extensively in their review, they are mostly directly involved in successful infection, not necessarily in the development of symptoms. Therefore, I find misleading the use of "Pathogenic mechanisms" term when speaking of effectors. I would suggest something along the lines of "Action mechanisms of effectors in plant-pathogen interaction".

2) I find that some sections are very skewed in their examples towards fungi. To better clarify for a reader that is new on the subject of plant-pathogen interaction, I'd suggest to say explicitly which mechanisms are found in both fungi and bacteria in each section, or which are unique to one of the two kingdoms. For example, cell-wall degrading enzymes can be found also in bacteria, but are only mentioned for fungi, and this can make it seem as if it's a strategy employed only by fungal pathogens.

3) Please check thoroughly the spelling of species name. There are some spelling mistakes (such as Fusium graminearum on line 221), and several style mistakes (such as having both genus and species name capitalized, for example in line 303) throughout the text.

Reviewer 2 Report

The paper “Pathogenic mechanisms of plant-pathogen effectors” is a review about an important topic in plant pathology. The text is well written, easy to read even if some acronyms are not explicit (or are only explained after their first introduction). The literature is comprehensive and up-to-date and I really appreciate figures. I have no particular comments to make, the article could be accepted in this form with the exception of some minimal integrations.

Here my specific comments:

General comment: the Authors never report viruses as pathogens. I think a comment on this is useful, also in relation to the role of viral proteins as elicitor recently proposed in recent years.

General comment: please check in the whole manuscript the use of abbreviation for the pathogens. Full name should be used when the genre was introduced the first time, then abbreviation should use. Authors seems not consistent with that (eg L298-306 Verticillium was never abbreviated even if previously reported, while other pathogen are abbreviated since first citation)

General comment: there is massive use of abbreviations for processes or molecules. However, Authors sometimes take the reader's knowledge of the abbreviations' meanings for granted. Sometimes the explanation of the acronym is given many phrases or chapters after its first use or repeated in differet chapter (eg. ROS). Please check the whole paper and be consistent. Some examples are reported as specific comments below.

L26: please define “PTI”

L28: please define “NLR”

L29: please define “ETI”

L88: please define “HR”

L111: please add some examples of “semi-biotrophic pathogens”

L123-148: In this part only bacteria are mentioned. However, there are important fungal pathogens with passive penetration via stomata. I think it is useful to add something about it.

L150-167: I suggest more data about Botrytis cinerea due to its importance for in-field/post-harvest disease in both herbaceous and woody plants.

L223: Fusarium graminearum

L277: please define “DAMPs”

L303: please check capital letters for “irregularis”

L503: please define “SA” if not done before

Fig.4: the legend is too long. Authors have to shorter it (no more than 20 lines) or report the description within the chapter, linking the various sentences to the figure.

L667: please define “JA” if not done before

L672: please define “BR”

Fig.5: same problem of Fig.4.

Reviewer 3 Report

Manuscript title: Pathogenic mechanisms of plant-pathogen effectors

Manuscript ID: ijms-1761685

Journal: IJMS, MDPI

The main objective of the current review article was to review recent advances made in the field of plant-pathogen interaction models and action mechanisms of phytopathogenic effectors. The subject is interesting and the manuscript was well-written. After reading the whole MS, I suggest the authors to mention in the introduction that the mechanism of action of any alternative control methods is divided to two main categories directly (on the pathogen itself) and indirectly (natural host resistance). There are several published articles which can help such as:

Journal of Fungi, 6, 179. https://doi.org/10.3390/jof6030179  

Scientia Horticulturae, 198: 86–90. http://dx.doi.org/10.1016/j.scienta.2015.11.013 

Acta Horticulturae, 1065:1593-1598. https://doi.org/10.17660/ActaHortic.2015.1065.203

I also suggest the authors to change the title to more attractive one. After considering the above mentioned note the review can be accepted in IJMS.

Reviewer 4 Report

Very nice review, the information have been published before in many reviews but there an effort did to clrify ideas in your work. The paper was well written but just verify in some places fungus (singular) and fungi (plural) and in the figuer 1 oomycetes to be equal to fungi and bacteria (used in plural).  Please revise the English for minor mistakes as in line 110, I don't understand why you wrote Plasmids? 

In general, the paper is very well and I recommend it for publication at its state.

Author Response

This manuscript is a resubmission of an earlier submission. The following is a list of the peer review reports and author responses from that submission.